# Lysine acetylation regulates the interaction between proteins and membranes

Alan K. Okada [1,8], Kazuki Teranishi[2,8], Mark R. Ambroso[2,8], Jose Mario Isas[2], Elena Vazquez-Sarandeses[3,4], Joo-Yeun Lee [5], Arthur Alves Melo[3,4], Priyatama Pandey[6], Daniel Merken[6], Leona Berndt[7], Michael Lammers[7], Oliver Daumke [3,4], Karen Chang[2,5], Ian S. Haworth [6] & Ralf Langen[2✉]

Lysine acetylation regulates the function of soluble proteins in vivo, yet it remains largely unexplored whether lysine acetylation regulates membrane protein function. Here, we use bioinformatics, biophysical analysis of recombinant proteins, live-cell fluorescent imaging and genetic manipulation of *Drosophila* to explore lysine acetylation in peripheral membrane proteins. Analysis of 50 peripheral membrane proteins harboring BAR, PX, C2, or EHD membrane-binding domains reveals that lysine acetylation predominates in membrane-interaction regions. Acetylation and acetylation-mimicking mutations in three test proteins, amphiphysin, EHD2, and synaptotagmin1, strongly reduce membrane binding affinity, attenuate membrane remodeling in vitro and alter subcellular localization. This effect is likely due to the loss of positive charge, which weakens interactions with negatively charged membranes. In *Drosophila*, acetylation-mimicking mutations of amphiphysin cause severe disruption of T-tubule organization and yield a flightless phenotype. Our data provide mechanistic insights into how lysine acetylation regulates membrane protein function, potentially impacting a plethora of membrane-related processes.

[1] Regions Hospital Department of Emergency Medicine, Saint Paul, MN 55101, USA. [2] Zilkha Neurogenetic Institute, Department of Physiology and Neuroscience, University of Southern California, Los Angeles, CA 90033, USA. [3] Max-Delbrück-Center for Molecular Medicine, Crystallography, Robert-Rössle-Straße 10, 13092 Berlin, Germany. [4] Institute of Chemistry and Biochemistry, Freie Universität Berlin, Takustraße 6, 14195 Berlin, Germany. [5] Neuroscience Graduate Program, University of Southern California, Los Angeles, CA, USA. [6] Department of Pharmacology and Pharmaceutical Sciences, University of Southern California, Los Angeles, CA 90089, USA. [7] Institute for Biochemistry, Synthetic and Structural Biochemistry, University of Greifswald, 17489 Greifswald, Germany. [8] These authors contributed equally: Alan K. Okada, Kazuki Teranishi, Mark R. Ambroso, Jose Mario Isas. ✉email: Langen@usc.edu

A myriad of cellular functions relies on the ability of proteins to interact with cellular membranes. Electrostatic interactions between proteins and membranes are fundamental to allowing these functions to take place[1]. Negatively-charged lipids such as phosphatidylserines and phosphatidylinositols (PIs) are ubiquitous constituents of intracellular leaflets of cellular membranes that commonly participate in protein interactions mediated by basic lysine and arginine residues[2]. Through such interactions, peripheral membrane proteins carry out a host of critical cellular activities including signaling, trafficking, and defining cell structure[2,3]. To do this, protein-membrane interactions help to localize protein machineries to specific sub-cellular locations, often reshaping membranes into functionally specific shapes necessary to carry out cellular functions. Consequently, the cell can perform highly coordinated tasks such as the synaptic vesicle cycle, formation of T-tubule networks in muscle, neutrophilic oxidative burst, endocytic processes and other vital functions[2,4–7]. For a cell to be appropriately responsive to its internal and external environment, these protein-membrane interactions must be performed in a manner tightly regulated in time and space.

In principle, such tight control of protein function can be executed via post-translational modifications (PTMs), which allow for rapid and robust regulation of large sets of proteins in vivo. There are many forms of PTMs, the most common being phosphorylation, N-glycosylation and lysine acetylation[8]. Not surprisingly, phosphorylation, the most well-studied of the PTMs, can exert influence over protein-membrane interactions by changing the charge potential of membrane-exposed residues[9–12]. Lysine acetylation, which we will refer to as acetylation, is the third most common form of PTM[8] and is well-known to regulate protein-DNA[13] and protein-protein[14,15] interactions. In contrast to phosphorylation, acetylation neutralizes basic residues by adding an acetyl group to lysine residues, yielding a neutral amide[16]. This activity in soluble proteins, is enzymatically controlled and can thus be regulated in time and space[17]. The biochemical, cellular and organismal consequences of this form of regulation are well-documented in the setting of protein-DNA and protein-protein interactions[15] but little is known about acetylation as a regulator of protein-membrane interactions.

Here, we investigate the role of acetylation in regulating the interaction between proteins and membranes using a bioinformatics approach to ascertain the degree to which four well-described families of membrane-binding domains, bin/amphiphysin/rvs (BAR), phox-homology (PX), C2, and Eps15-homology domain containing proteins (EHDs) are acetylated within or outside regions that interface with membranes. We further characterize the functional consequences of acetylation on protein-membrane interactions using biophysical analysis of recombinant proteins as well as cellular and animal model approaches that utilize acetylation-mimicking mutations to study candidates from our bioinformatics analysis. Our data reveal that acetylation is significantly more prevalent in regions directly involved in membrane interactions compared to regions in the same domains that do not interact with membranes. Acetylation as well as mimicking acetylation in candidate proteins strongly reduces membrane binding and alters membrane remodeling. In cell and animal models, this leads to a sub-cellular redistribution of the proteins studied, with significant tissue level and behavioral consequences. Taken together, our data suggest that acetylation can regulate protein-membrane interactions and membrane protein function.

## Results

**Acetylated lysines are strongly enriched in membrane-interaction regions.** Peripheral membrane proteins interact with membranes via domains specially designed to contact membranes. For acetylation to play a widespread role in controlling protein-membrane interaction and function, one might expect acetylation sites to be specifically targeted to the regions of membrane-binding domains that participate in membrane interaction. To test this notion, we developed a bioinformatics approach that combines the known locations of post-translational lysine acetylation sites with high-resolution structural information of membrane-binding domains and their membrane-interaction region. We chose domains from four structurally well-characterized families of peripheral membrane-binding proteins BAR[2,9,18–30], PX[3,6,27,31–40], C2[1,3,4,41–64], and EHD[65–67], for which acetylation data were available from Phosphosite.org[68]. We compiled empirical data from the literature for structurally well-characterized members of each domain family regarding the specific surfaces directly involved in interactions with membranes. Regions that interact with membranes were defined as membrane-interaction regions (MIRs), whereas all other regions were defined as non-binding regions (NBRs). In this way, structural templates with experimentally defined MIRs and NBRs were generated for each type of domain. We next performed structural or sequence-based alignments between the templates and the other domain family members (schematically illustrated in Supplemental Fig. 1). When available, structural alignment was chosen over sequence alignment. Sequence-based alignments were performed for those proteins for which no high-resolution structural data were available. Lysine acetylation data from Phosphosite.org were then integrated into both types of alignments in order to ascertain whether acetylation is enriched within MIRs.

As summarized in Fig. 1, we find that acetylation localizes predominantly to the MIRs of membrane-binding domains. Of all domains tested, 82% exhibit a higher degree of acetylation in their MIRs than in their NBRs (Fig. 1A). Remarkably, 60% of the domains analyzed exhibit acetylation exclusively in their MIRs (Fig. 1A). Among the four domain families, the results range from 75% to 92% (Fig. 1B) and the detailed breakdown for each family member is provided in Supplemental Figs. 2–5. These findings were relatively insensitive to the precise assignment of the MIR, as expanding and contracting the definitions by 1 to 2 amino acids did not change the overall trends (Supplemental Fig. 6).

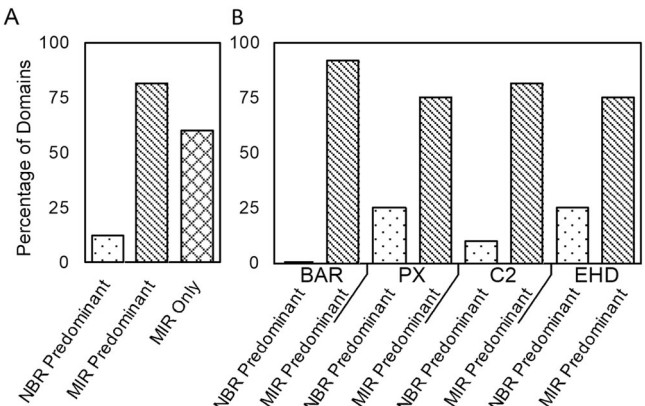

**Fig. 1 Acetylation is found predominantly in membrane-interaction regions.** Fraction of domains with the majority of acetylation in the membrane-interaction region (MIR) and non-binding region (NBR) or only in the MIR (MIR Only) for the entire data set (**A**). The breakdown of the data grouped by individual domain families, BAR, PX, C2 and EHD is given in **B**. Three cases total were found with equal number of acetylation in the MIR and NBR. These rare ties were not plotted for simplicity. For a detailed protein-by-protein list, see supplemental Figs. 2–5. 12 BAR domains, 12 PX domains, 22 C2 domains, and 4 EHD domains were included in the analysis. Source data are provided as a Source Data file.

In order to better understand why acetylation is selective for MIRs, we first determined the acetylation frequencies for MIRs and NBRS by normalizing for the number of amino acids in each region. Overall, we observed a 4.7-fold enhancement of acetylation within MIRs relative to NBRs, which is shown in Table 1, along with a detailed breakdown for each domain family. The substantial enrichment of acetylation within MIRs is not just a consequence of the greater prevalence of lysines within the MIRs compared to NBRs (Supplemental Table 1) as an increased likelihood of acetylation within MIRs is still observed when the acetylation frequency is normalized by the number of lysines in each domain (Table 1).

Further study of our structural alignments reveals that the distribution of acetylation within membrane-binding regions tends to cluster around key membrane-binding sites. In the case of the BAR domains, we see acetylated lysines confined primarily to the curvature-generating, concave surface, where 44/45 are oriented along or into the membrane interfacial region[9,19,21–24,26,28,69–73] (Fig. 2a–c). In addition, we find acetylation within the N-terminal H0 helices of N-BARs, which are often not resolved in crystal structures, but which form in the presence of membranes[9,23,25,26] (Supplemental Fig. 2c). Likewise, acetylated lysines within the MIRs of PX domains cluster primarily around the phosphatidylinositol binding pocket (Fig. 2d arrow) which provides specificity and a driving force for interactions with PI3-P containing membranes[31,33,36,39,49,74]. Also, in the case of C2 domains, we observe lysine acetylation in regions critical for membrane binding, including both the $Ca^{2+}$-dependent ($Ca^{2+}$ binding loops 1-3) and $Ca^{2+}$-independent (polybasic) membrane-binding regions (Fig. 2e arrowhead and arrow, respectively). The acetylated lysines within the polybasic region are clustered at the core of the membrane-binding groove along the two β-strands, where they are perfectly situated for membrane interaction. Similarly, a pattern of acetylation emerged among the EHD family members wherein acetylation is clustered into the so-called tip region, which is the primary membrane binding site of the EHD helical domain[65–67] (Fig. 2f arrow). Not only did we find acetylation concentrated within the MIRs of all four types of domains, but we also consistently found acetylated lysines clustered in and around structural motifs critical for protein-membrane interaction. The conserved nature of this preferential distribution of acetylation, especially with respect to the specific distribution of acetylation within MIRs, is consistent with acetylation playing an important role in controlling membrane protein function that is generalizable across many membrane-binding domains.

**Acetylation and acetylation mimetics of EHD2 decrease membrane affinity, binding, catalytic activity and alter its membrane remodeling.** Having established a significant preference for acetylation sites within MIRs, we next sought to understand the functional consequences of such modifications. The neutralization of a lysine positive charge by acetylation could function as a switch by reducing interactions with negatively charged, intracellular membranes. To ascertain whether acetylation modulates protein-membrane interactions, we investigated three well-characterized peripheral membrane protein candidates with membrane-binding domains from the EHD, BAR, and C2 families, EHD2[65–67], amphiphysin[23,26], and synaptotagmin1[4,43,44,75–78], respectively. To recapitulate acetylation in a manner that could also be employed in cell and in vivo models, lysine to glutamine mutants were generated and tested in vitro. This is a common method for mimicking acetylation that, like acetylation, results in loss of the positive lysine charge in lieu of an amide bond[79]. For selected proteins, we also produced variants that were specifically acetylated on a single lysine.

As the first test case, we studied EHD2, a member of the EHD family, which binds and tubulates membranes[65]. EHD2 is well-characterized and contains 4 lysines that have been found to be acetylated, all of which reside in the MIR (Supplemental Fig. 5). When EHD2 binds membranes, K324, located in the primary

**Table 1 Prevalence of lysine acetylation in membrane-interaction regions and non-binding regions of BAR, PX, C2 and EHD membrane-binding domains.**

| Domain | $\%K_{AC}$ | | | $\%\frac{K_{AC}}{K}$ | | |
|---|---|---|---|---|---|---|
| | MIR | NBR | Ratio | MIR | NBR | Ratio |
| BAR | 3.20 | 0.61 | 5.21 | 20.5 | 9.8 | 2.1 |
| PX | 2.82 | 0.57 | 4.92 | 26.3 | 9.3 | 2.8 |
| C2 | 3.30 | 0.67 | 4.90 | 22.9 | 10.5 | 2.2 |
| EHD | 5.85 | 1.84 | 3.18 | 28.9 | 27.8 | 1.0 |
| Average | 3.30 | 0.71 | 4.65 | 22.8 | 11.3 | 2.0 |

MIR, membrane-interaction region; NBR, non-binding region; Ratio, the value for MIR divided by NBR for $\%K_{AC}$ or $\%\frac{K_{AC}}{K}$; $\%K_{AC}$, number of acetylated lysine residues normalized by the total number of amino acids in the region; $\%\frac{K_{AC}}{K}$, number of acetylated lysine residues normalized by the number of lysine residues in the region. 12 BAR, 12 PX, 22 C2 and 4 EHD domains were used in the analysis. Source data are provided as a Source Data file.

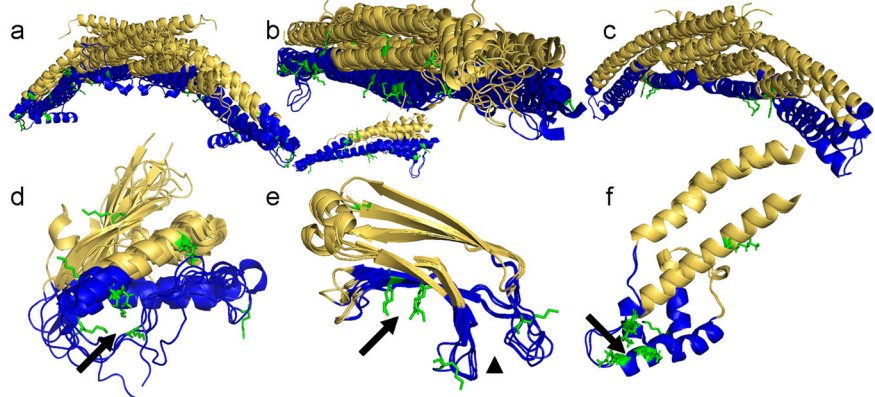

**Fig. 2 Acetylation within domains is primarily localized to membrane-interaction regions.** Overlays of all structures from **a** N-BAR, **b** F-BAR (top - central region, below - tip region), **c** PX-BAR, **d** PX (arrow indicates PI binding pocket), **e** C2 (arrow indicates $Ca^{2+}$-independent binding region; arrowhead indicates $Ca^{2+}$ binding loops 1–3), and **f** EHD helical domains (arrow indicates membrane-interacting tip region of the α9 helical domain as defined for EHD2). For a detailed list of PDB structures, see supplementary excel and Supplemental Figs. 2–5. MIRs are shown as ribbons in blue. NBRs are shown as ribbons in gold. Acetylated lysines are shown as sticks in green.

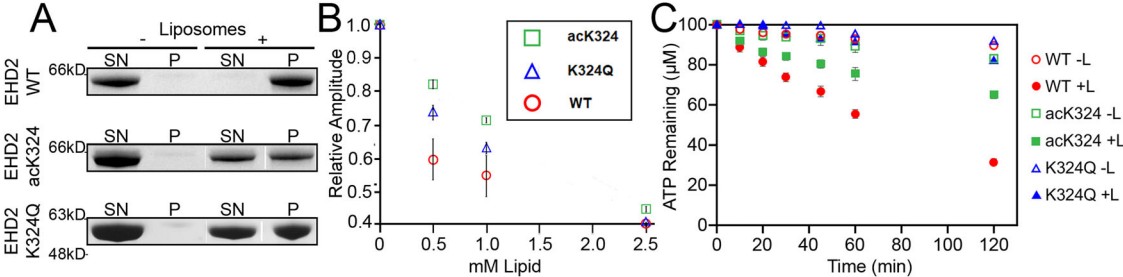

**Fig. 3 Acetylation reduces EHD2 membrane binding and membrane binding affinity. A** Co-sedimentation assays of EHD2-WT, EHD2-acK324 or EHD2-K324Q in the absence or presence of liposomes. SN, supernatant; P, pellet fraction. **B** mean of EPR spectral amplitudes plotted as a function of lipid concentration from EHD2-WT (red circle), EHD2-acK324 (green square) and EHD2-K324Q (blue triangle) spin-labeled at position 321. **C** ATP hydrolysis by EHD2-WT (red circle), EHD2-acK324 (green square) and EHD2-K324Q (blue triangle) in the presence (filled marker) or absence (empty marker) of liposomes was determined by an HPLC-based method. Error bars represent the range (s.e.m.) of 3–6 independent experiments. Co-sedimentation was performed three times for EHD2-acK324 and twice for EHD2-K324Q with wild type controls in each independent experiment. Source data are provided as a Source Data file.

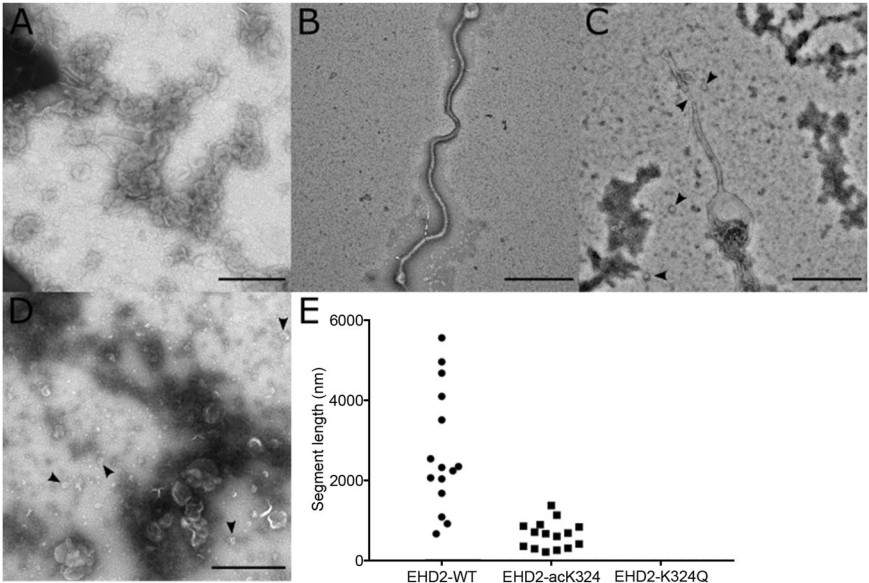

**Fig. 4 Acetylation and acetylation mimetics alter membrane remodeling of EHD2. A–D** TEM of liposomes incubated without protein (**A**), or with EHD2-WT (**B**), EHD2-acK324 (**C**) or EHD2-K324Q (**D**). Black arrowheads in **C** and **D** indicate detached particles ~30–50 nm in size, consistent with small vesicular structures. **E** Length of protein-decorated lipid tubule segments. Scale bars = 500 nm. Micrographs were repeated three times each in three independent assays. Source data are provided as a Source Data file.

membrane-interaction region, penetrates into the bilayer[66,67]. This membrane binding enhances EHD2's ATPase activity ~8-fold[65] and facilitates membrane remodeling. In order to determine whether lysine acetylation affects EHD2's membrane binding and function, we used two different approaches. First, we site-specifically incorporated the non-canonical amino acid $N$-(ε)-acetyl-L-lysine to the K324 acetylation site[36,61] (EHD2-acK324), using genetic code expansion[80,81]. Second, we mutated K324 to glutamine in order to mimic acetylation (EHD2-K324Q), a technique that translates readily from in vitro, to cellular and organismal level genetic manipulation.

We then analyzed membrane binding in co-sedimentation assays with liposomes composed of a lipid extract from cow brain (Folch). Whereas almost all EHD2-WT co-sedimented with the liposomes, this fraction decreased to 48% and 43% for EHD2-acK324 and EHD2-K324Q, respectively (Fig. 3A). These results show that both acetylation of K324 and acetylation mimicking glutamine mutation reduce membrane-binding. We next employed a previously developed electron paramagnetic resonance (EPR)-based method that measures membrane binding during lipid titration via

corresponding amplitude changes[9]. A right shift in the titration curve of EHD2-acK324 and EHD2-K324Q relative to EHD2-WT was evident, demonstrating that both acetylation and acetylation-mimicking mutations attenuate protein-membrane binding affinity (Fig. 3B). To evaluate whether the change in membrane-binding affinity leads to functional changes, we assayed EHD2 catalytic ATPase activity in EHD2-acK324 and EHD2-K324Q. Consistent with a reduction in membrane binding, both EHD2-acK324 and EHD2-K324Q conferred a significant decrease in the stimulated ATPase activity compared to EHD2-WT. The basal ATPase activities of all EHD2 variants without lipid were in a similar range (Fig. 3C). These data establish that acetylation and mimicking acetylation in EHD2 yield similar effects, reducing protein-membrane interaction.

To determine whether this modification affects EHD2 membrane remodeling ability, we employed transmission electron microscopy (TEM) to image EHD2 at different concentrations in the presence of membranes. At protein concentrations of 7.5 μM, EHD2-WT showed strong membrane remodeling, giving rise to tubules which frequently were several microns long (Fig. 4B). In contrast, the

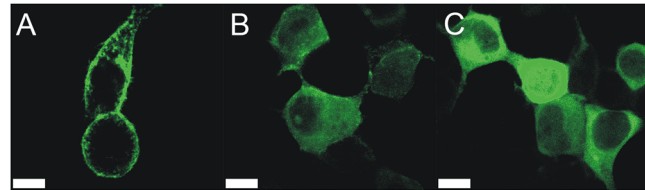

**Fig. 5 Mimicking acetylation inhibits EHD2 distribution to the plasma membrane, yielding a diffuse cytoplasmic distribution. A–C** Fluorescent confocal microscopy of HeLa cells expressing GFP-EHD2-WT (**A**), GFP-EHD2-K324Q (**B**) or GFP-EHD2-324Q328Q (**C**). Scale bars: white bar = 10 μm. Representative images taken from 3 independent experiments.

acetylated protein lost its ability to generate long and interconnected membrane tubules, instead forming shorter, predominantly sub-micron long membrane tubules (Fig. 4C, E) as well as more frequent, small vesicular structures, likely indicating vesiculation (Fig. 4c, black arrowheads). This behavior is analogous to what has previously been reported for a phosphomimetic mutant (S75D) of endophilin[9]. EHD2 K324Q had an even more pronounced effect, as mainly vesicular structures with a range of sizes could be seen (Fig. 4D). It has previously been suggested that tubulation requires a minimal protein to lipid ratio[9,23,82–84]. In fact, when we used higher concentrations (10 μM), the differences between EHD2-WT and the acetylated protein were less pronounced, as both were able to form long tubules. However, the acetylated protein still appeared to have a slightly higher propensity for forming more vesicular structures. For EHD2-K324Q, mostly vesicular structures were again seen, in addition to very rare tubules (Supplemental Fig. 7). While the effects of acetylation and acetylation mimicking were not fully identical in all cases, both had similar pronounced effects relative to EHD2-WT, decreasing the membrane affinity of EHD2, decreasing protein-membrane binding, functionally reducing its catalytic activity and altering its membrane remodeling capacity.

**Acetylation mimetics alter EHD2 subcellular distribution.** In HeLa cells, N-terminally GFP-tagged EHD2 (GFP-EHD2-WT) is distributed to the cell membrane and the adjacent cytoplasmic region[65] (Fig. 5A). Expression of the single K to Q mutant, GFP-EHD2-K324Q, and the double mutant containing K to Q mutations at two membrane binding residues, 324 and 328, GFP-EHD2-324Q328Q[14,68] in HeLa cells resulted in a diffuse cytoplasmic distribution (Fig. 5B, C). These findings further underscore the potent effect that acetylation mimetics can have on binding to cellular membranes.

**Acetylation and acetylation mimetics block membrane remodeling of amphiphysin.** The N-terminus of the amphiphysin BAR domain is important for inducing membrane curvature and has acetylation sites at positions 5 and 15[23,26,68,85]. To test the effects of acetylation on amphiphysin, we used genetic code expansion as before to acetylate K15 (Amph-acK15). We also mutated K5 and K15 to glutamines (Amph-5Q15Q). Using TEM, we first studied the effects of acetylation and mimicking acetylation in *Drosophila* amphiphysin on lipid remodeling with Amph-acK15 and Amph-5Q15Q. Incubation of wild type *Drosophila* amphiphysin (Amph-WT) with a lipid composition designed to mimic cellular membranes resulted in near complete tubulation of liposomes. Liposomes incubated with acetylated amphiphysin or the acetylation mimic, however, resembled micrographs of lipid alone (Fig. 6A–D). To test whether the loss of tubulating phenotype was due to the changes in charge or side chain structure, we replaced positions 5 and 15 with positively charged arginine residues (Amph-5R15R) and found that tubulation was restored (Fig. 6E), similar to Amph-WT. Thus,

acetylation and acetylation-mimicking mutations significantly impacted amphiphysin's tubulation and membrane remodeling abilities while the charge-maintaining arginine mutations resulted in functional protein.

**Acetylation mimetics change subcellular distribution of amphiphysin and decrease membrane-binding affinity.** In order to test the cellular effects of the acetylation at K5 and K15, we expressed C-terminally green-fluorescent protein labeled amphiphysin wild type (WT-Amph-GFP), K5Q K15Q double mutant (5Q15Q-Amph-GFP), or K15Q single mutant (15Q-Amph-GFP) in COS-7 cells. As in prior studies[26], we found WT-Amph-GFP to be associated with tubular networks (Fig. 7A). Conversely, neither 15Q-Amph-GFP nor 5Q15Q-Amph-GFP formed tubular networks (Fig. 7B, C). This change is consistent with the reduced membrane-binding and tubulation activity identified by biophysical analysis. Having seen robust effects on tubulation with both acetylation and acetylation-mimicking mutations along with the striking change in cellular phenotype, we confirmed that, as in the case of EHD2, the inhibition of tubulation was caused by a reduction in binding affinity (Supplemental Fig. 8).

**Mimicking acetylation in synaptotagmin1 reduces membrane-binding affinity.** As a final test case, we examined whether the acetylation-mimicking K237Q mutant in the C2A membrane-binding region of the C2 domain-containing protein synaptotagmin1 (Syt1-K237Q) could alter $Ca^{2+}$-dependent synaptotagmin1 membrane interactions. Lysines within the membrane binding loops of synaptotagmin1 are known to perform critical functions in the interaction between the protein and its target membrane[58–64]. K237 is located in the third calcium binding loop and involved in $Ca^{2+}$-dependent phospholipid interactions[4]. This mutation also led to a right-shift in the lipid titration curve for Syt1-K237Q compared to wild type controls, indicating reduced $Ca^{2+}$-dependent membrane binding affinity (Supplemental Fig. 9).

**Mimicking acetylation in amphiphysin leads to destabilization of T-tubule networks and loss of flight in Drosophila melanogaster.** Having established that acetylation mimetics in the membrane-interaction region of different proteins reliably affect membrane interaction in vitro and in cells, we sought to further validate our findings using an in vivo system. Amphiphysin stabilizes the T-tubule network in *Drosophila melanogaster* muscle tissue[5]. To test whether mimicking acetylation of amphiphysin alters its function in vivo, we generated *Amph-WT* and *Amph-5Q15Q* transgenic flies in an amphiphysin-null background (*amph[26]*; hereafter referred to as *amph[null]*). The T-tubule network in the adult indirect flight muscles was examined using an antibody against Discs large (Dlg), a marker for T-tubules[5,86]. Normal flies exhibited a characteristic pattern of Dlg staining, whereas *amph[null]* showed disorganized and reduced Dlg staining, indicating defective T-tubule formation (Fig. 8A). Expression of *Amph-WT* rescued T-tubule formation, while *Amph-5Q15Q* yielded severely disrupted T-tubules similar to *amph[null]*, despite comparable expression levels (Fig. 8A, B).

Consistent with T-tubule disruption in muscles, *amph[null]* flies displayed a flight defect (Fig. 8C). This defect was ameliorated by *Amph-WT* expression, whereas expression of *Amph-5Q15Q* did not rescue the flight deficit of *amph[null]* flies (Fig. 8C). The T-tubule degeneration and the flightless phenotypes of the acetylation mimicking mutant are in agreement with the biochemical and cell data, which reveal a strongly reduced ability to stabilize tubular structures.

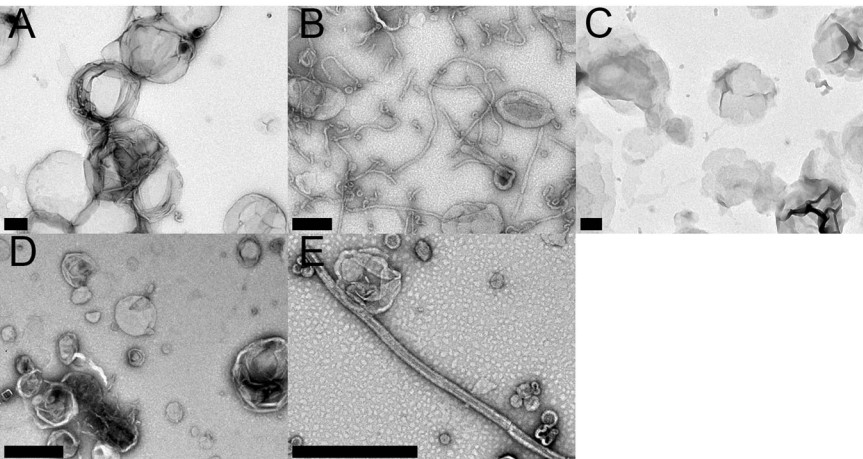

**Fig. 6 Acetylation and acetylation-mimicking mutations inhibit Amphiphysin membrane remodeling. A–E** TEM of liposomes incubated in the absence of protein (**A**), or with Amph-WT (**B**), Amph-acK15 (**C**), Amph-5Q15Q (**D**), or Amph-5R15R (**E**). Micrographs were repeated three times each in three independent assays. Scale bar = 200 nm.

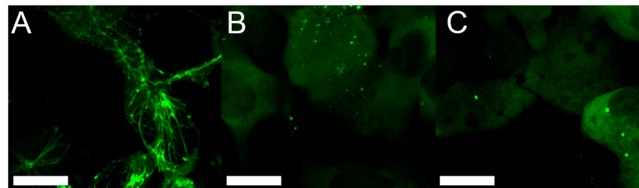

**Fig. 7 Mimicking acetylation inhibits amphiphysin's subcellular tubular distribution, yielding a diffuse cytoplasmic distribution. A–C** Fluorescent confocal microscopy of COS-7 expressing WT-Amph-GFP (**A**) 15Q-Amph-GFP (**B**) or 5Q15Q-Amph-GFP (**C**). Scale bar = 10 µm. Representative images taken from 3 independent experiments.

## Discussion

Lysine acetylation is an important and common regulatory mechanism for cellular homeostasis[8]. Here, we present several lines of evidence in support of the idea that acetylation regulates the functions of multiple families of peripheral membrane proteins. Acetylation is significantly enhanced in membrane-binding regions, where it often directly localizes to critical membrane binding pockets (Fig. 2), in a perfect position to modulate membrane interactions. Acetylation and acetylation mimetics strongly affected membrane interaction of all candidate proteins studied here, causing reduced membrane affinity and, in the case of amphiphysin and EHD2, altering membrane remodeling. In cells, mimicking even a single acetylation event within the MIRs reduced binding affinity to membranes leading to cytoplasmic dispersion. The highly potent effect of acetylation mimetics on interactions with cellular membranes was further underscored by our in vivo studies, where Amph-5Q15Q caused a dramatic effacement of the T-tubule network with a corresponding loss of *Drosophila* flight behavior. The notion that these effects are largely caused by charge neutralization of key lysine residues is supported by experiments in which the membrane binding and remodeling ability of amphiphysin was rescued by introduction of arginine residues. The lysine to arginine mutations significantly alter sidechain geometry, but they maintain the positive charge, indicating that charge is important for membrane binding and remodeling. A charge neutralization mechanism has previously been identified as an important factor by which lysine acetylation modulates protein-nucleic acid and protein-protein interactions[15]. Our results support the idea that the same mechanism extends to protein-membrane interaction (Fig. 9).

As acetylation is strongly enhanced throughout key membrane-interaction regions in a variety of protein families, we suspect acetylation could control a vast range of membrane-related, cellular processes (Fig. 9). The BAR and EHD families of proteins have been implicated in membrane trafficking and remodeling events, including filopodia and lamellipodia formation, endocytosis, endosome-Golgi transport, macroendocytosis and caveolae stabilization[7,20,87–90]. Based on the pronounced impacts on membrane interaction as well as membrane remodeling observed in the present study, all of these processes could be potentially affected by acetylation. Likewise, the ability to control membrane-binding activity of C2 domains via acetylation could allow the cell to further regulate $Ca^{2+}$-dependent membrane trafficking and signal transduction events[4]. The set of functions for which PX domains are necessary are even more varied, ranging from oxidative burst, to membrane remodeling, signaling, and a host of motor and enzymatic functions[6,31,91–93]. While the protein families investigated here already span a wide range of membrane-related cellular events, it is quite likely that acetylation is also utilized to regulate many additional families of peripheral membrane proteins and, thereby, many other additional cellular processes. Here, we found that acetylation of peripheral membrane proteins functioned as an attenuator of membrane interaction, strongly affecting membrane remodeling (Fig. 9). While we expect acetylation to attenuate membrane interaction in many additional membrane proteins, one could also envision certain scenarios where acetylation may have more diverse effects. Acetylation could perhaps result in a more graded response by altering the mode of membrane interaction. For example, acetylation could alter the immersion depth of amphipathic helices as seen in the case of phosphorylation[9]. Acetylation might also alter lipid specificity. For instance, in proteins interacting with highly negatively charged lipids (e.g. PI with multiple phosphates), the loss of positive charge via acetylation may shift the specificity to a less phosphorylated PI or even neutral lipid. While this would likely have effects on membrane localization, it may also affect enzymatic activity. In fact, the loss of ATPase activity we see with EHD2-acK324 in our studies presented here is one such example. Similar mechanisms might also apply to PTEN, which is acetylated in its catalytic cleft at K125/K128 where acetylation attenuates enzymatic activity[94]. Interestingly, acetylation exists on the membrane-binding surface of the phosphatase domain at K163/K164 where alanine mutations appear to reduce membrane-binding[95]. Two manuscripts report

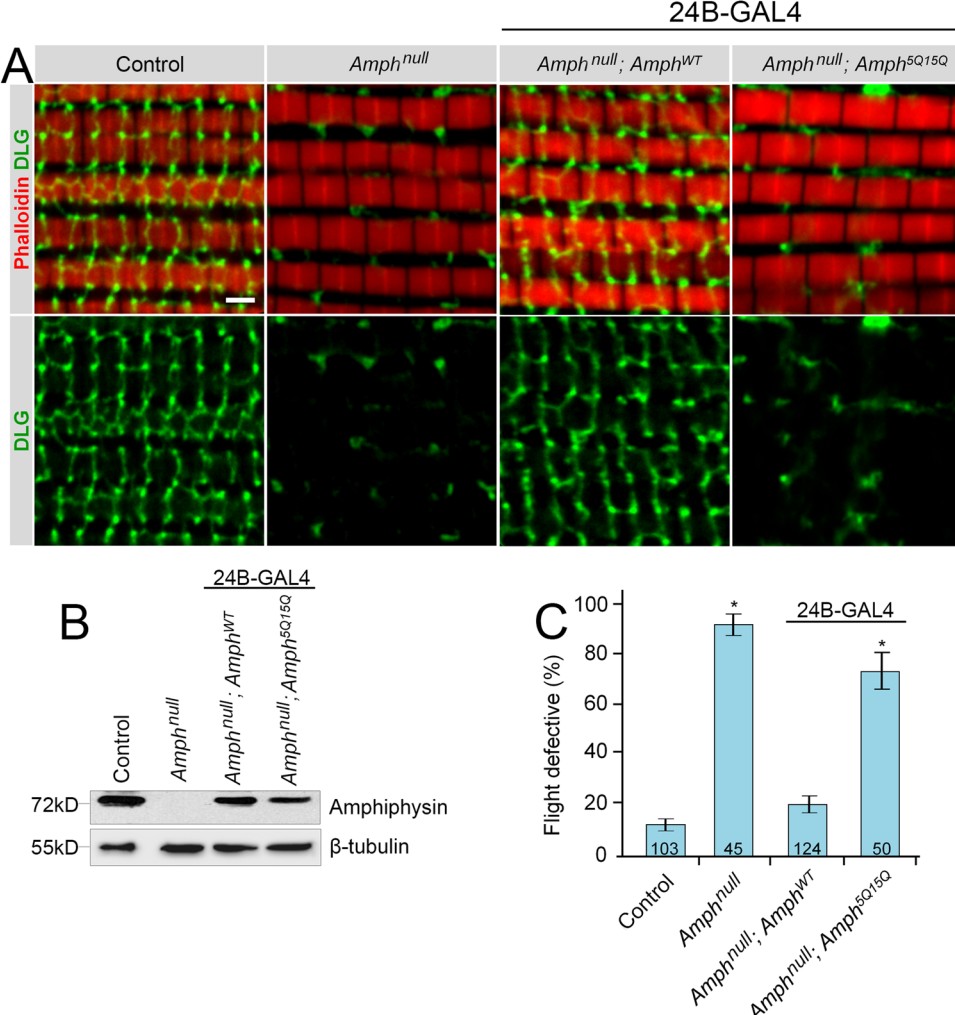

**Fig. 8 Amph-5Q15Q flies display defective T-tubule organization and a flight deficit. A** Representative images of adult IFM stained for actin (phalloidin, red) and DLG (green), with *Amph-WT* or *Amph-5Q15Q* expression driven by 24B-GAL4 over an *Amphiphysin-null* (*Amph^null^*) genetic background from >4 independent experiments per genotype. **B** Western blot demonstrating levels of transgene expression detected using an antibody against the SH-3 domain of Amphiphysin. **C** Percentage of flies that are flight defective. "*n*" is reported at the base of each condition where, *n* = number of flies tested. For control, *Amph^null^*, *Amph^WT^*, *Amph^5Q15Q^*, *n* = 103, 45, 124, 50 animals, respectively. Error bars represent ±1 SD from ≥5 experiments. Scale bar = 2 μm. *$p$-value < 0.05. $P$-values relative to control are *Amph^null^* $p = 1.9 \times 10^{-7}$, *Amph^null^*;*Amph^5Q15Q^* $p = 1.3 \times 10^{-5}$. Western blots performed for a minimum of 3 independent experiments. Source data are provided as a Source Data file.

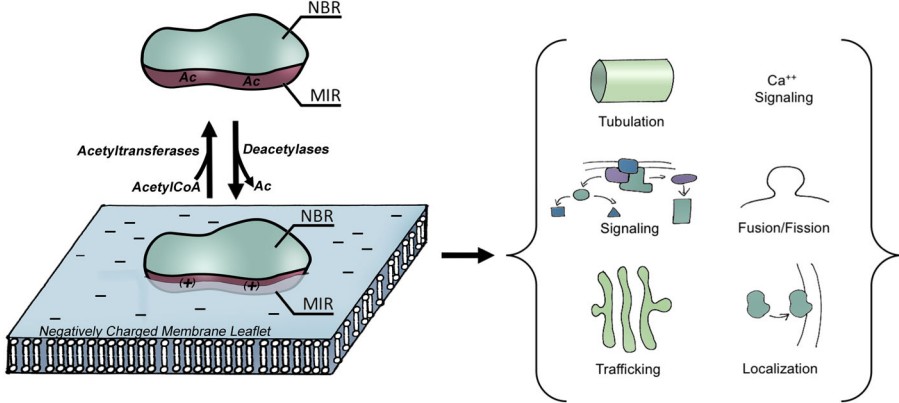

**Fig. 9 Proposed model of effects of lysine acetylation on protein-membrane interactions.** Peripheral membrane proteins carry out numerous vital cellular functions (right) via interactions with cellular membranes (bottom left). Membrane-interaction region (MIR) and non-binding region (NBR) are colored mauve and teal, respectively. Acetylation alters protein-membrane interactions by neutralizing membrane-interacting lysines of peripheral membrane proteins (top left). The plus signs indicate positively charged lysines, whereas Ac indicates acetylated, neutral lysines.

acetylation within putative membrane binding regions not belonging to any known family of membrane binding domains[96,97]. Moreover, two reports on PH domain containing proteins indicate that acetylation causes opposing effects on membrane localization in cells (either increasing or decreasing)[98,99]. While the differential effects of acetylation seen in the two PH domains could indicate selective changes in specificity, as we suggested above, PH domains also participate in protein-protein interactions[100], which could contribute to cellular localization and be affected by acetylation. This feature makes PH domains less ideal candidates for studying the effects of acetylation on protein-membrane interaction. Our data, however, consistently show that acetylation disrupts membrane binding in membrane-binding domain families that, to our knowledge, have never been analyzed for the effect of acetylation in protein-membrane interaction. This consistency may, in part, be attributable to the fact that all four families have high affinities for negatively charged membranes.

Our data show that glutamines are effective mimetics of lysine acetylation for amphiphysin function insofar as they replicate charge neutralization. The success of this approach allowed us to genetically encode the effect of acetylation in cellular and animal studies by applying Lys to Gln mutations. As seen in Fig. 4, however, acetylation and glutamine acetylation mimetics led to similar but not identical effects on EHD2. While charge neutralization is maintained, the glutamine side chain is less hydrophobic and less bulky compared to acetylated lysine. As acetylated lysine has a steric and hydrophobic component that is not mimicked by a glutamine mutation of lysine[101–103], it is likely that these differences affect membrane interaction properties such as depth of insertion into the bilayer compared to true acetylation. Thus, for EHD2, our data indicate that charge neutralization is a major factor in the function of acetylation, but does not comprise the entire mechanism, whereas for amphiphysin, charge neutralization may be the dominant driver.

It should also be noted that we focused solely on peripheral membrane proteins, however, transmembrane proteins often have lysine residues flanking their transmembrane helices and contain soluble domains that are located in an aqueous environment[104]. In fact, this feature may play a significant role for mitochondrial transmembrane proteins, which exist in an environment where acetylCoA (the substrate of acetylation) levels fluctuate as a function of metabolic state[105]. Thus, it may well be possible that principles uncovered here apply to transmembrane proteins as well.

Much has been learned about the acetyltransferase and deacetylase enzymes that regulate protein-DNA and protein-protein interactions[13–15]. Some of the same enzymes may also be involved in controlling protein-membrane interactions. If acetylation occurs in the cytosol this could clearly modulate the membrane binding affinity. However, since many lysine residues are in membrane proximity, effective acetylation of already membrane-bound proteins may require a set of enzymes that themselves are membrane-anchored in order to access the lysine residues that are in membrane proximity. Consistent with this notion, localization of acetyltransferases to the cytoplasm and deacetylases to cellular membranes has been observed[106–108]. Efforts to uncover the enzymes and the biochemical signaling cascades that they are a part of, are likely to yield a better understanding of a wide range of membrane-processes and ultimately it may even provide a new set of pharmacological targets for drug discovery.

## Methods

**Reagents**. 1-palmitoyl-2-oleoyl-*sn*-glycero-3-phospho-L-serine (POPS), 1-palmitoyl-2-oleoyl-*sn*-glycero-3-phosphocholine (POPC), 1-palmitoyl-2-oleoyl-*sn*-glycero-3-[phospho-RAC-(1-glycerol)] (POPG), and 1-palmitoyl-2-oleoyl-*sn*-glycero-3-phosphoethanolamine (POPE) were obtained from Avanti Polar Lipids (Alabaster, AL). Bovine total brain lipids, Folch fraction I was obtained from Sigma-Aldrich (St. Louis, MO). 1-oxyl-2,2,5,5 tetramethyl-Δ3-pyrroline-3-methylmethanethiosulfonate (MTSL) was purchased from Toronto Research Chemicals (Toronto, Ontario, Canada).

*Bioinformatics analysis of protein acetylation*. Acetylation data were acquired from www.phosphosite.org[68] between December 2016 to September of 2017. Structure files (www.rcsb.org) for membrane-binding domain families were downloaded in bulk. Scripts were generated to sort these.pdb files into homology clusters based on a structural homology algorithm generated in-house. Family members with less than 50% sequence identity were excluded from the analysis.

Determination of lysine acetylation frequency in membrane-binding and non-binding regions was performed as follows. Based on the availability of defined crystal structures and high resolution experimental data on membrane binding in the literature, proteins were chosen to serve as templates for structural and sequence comparisons within domain families. The domain ranges of the templates were defined based on UNIPROT (uniprot.org) definitions. In the case of BAR domains, N-terminal regions containing membrane-embedded helices were also included within the domains. In the case of EH domain proteins, domain ranges and terminology were defined according to the pioneering work of Daumke et al.[65]. The MIRs and NBRs of templates were defined by mapping the experimental data gathered from the literature onto the crystal structure using PyMOL. Charge density mapping was also used to help visualize and validate definitions determined from empirical data.

For domain family members with available crystal structures, but without empirically well-defined membrane-interaction regions, structural alignments were made using PyMOL (PyMOL Molecular Graphic System, Version 1.7.4 Schrödinger, LLC) against a template from which the domain range and MIR definitions were determined. For BAR domains lacking N-terminal helices in their pdb structures, MPEx[109] was used to define the helix for comparisons. For F-BAR a number of domains display a kink that rotates the tip region. Alignments in such cases were performed against regions comparable to the template on either side of the kink. The assignments were further verified by electrostatic potential mapping and validation with available literature.

For proteins lacking crystal structures, a sequence alignment method was used. MIRs and NBRs were defined from template domains and ensembles of protein sequences aligned using ClustalOmega (https://www.ebi.ac.uk/Tools/msa/clustalo/)[110]. Colorations for figures were generated using ESPript (PMID: 24753421 http://espript.ibcp.fr)[111] to highlight the highly conserved residues among each domain family. To avoid aligning non-functional C2 domains[112], we applied a threshold similarity score cutoff of <50% similarity, relative to the C2 templates, to exclude degenerate domains. Thus, for each C2 family member, the domain as defined by UNIPROT was individually aligned with both C2A and C2B of Syt1 using the EMBOSS NEEDLE protein alignment function (https://www.ebi.ac.uk/Tools/psa/emboss_needle/) and clustered with either C2A or C2B based upon similarity scores. Each set was then aligned using ClustalOmega.

Acetylation data were obtained from Phosphosite.org[68]. We generated a script to assign those sites within the defined MIR or NBR. Acetylated lysines in the MIR or NBR were tallied and normalized to the total number of amino acids or lysines in the respective regions. The code for these processes are available in our supplementary material. As a control to test the effects of our MIR assignments, we expanded and contracted the MIR definitions by two residues and repeated our analysis.

*Protein expression, DNA constructs and mutagenesis*. Plasmids and cDNA encoding rat synaptotagmin1 (a.a. 80–421), and *Drosophila* amphiphysin (a.a. 1–244) were the generous gifts from Drs. Greg Schiavo and Harvey McMahon respectively.

Recombinant expression and purification of proteins, including mouse EHD2 (a.a. 1–543) and all mutants were performed as previously described[23,26,67,113]. Briefly, proteins were expressed in *E. coli* BL21 (DE3) Rosetta (New England Biolabs). EHD2 and amphiphysin were purified using nickel-nitrilo-triacetic acid–agarose, followed by gel filtration with a Superdex 200 column. For amphiphysin remaining impurities were removed using mono S cation exchange chromatography with a low salt buffer A (20 mM HEPES pH 7.4, 1 mM dithiothreitol (DTT)) and elution buffer B (20 mM HEPES pH 7.4, 2 M NaCl and 1 mM DTT). Synaptotagmin1 was purified by immobilizing the protein on glutathione-agarose followed by extensive washing. Synaptotagmin1 was eluted off the beads by thrombin cleavage of the GST tag from the protein. Protease and other impurities were removed using a mono Q column (GE). Protein concentrations were determined by UV-absorbance at 280 nm. The purified samples were flash frozen and stored at -80°C.

The site-specific incorporation of *N*-(ε)-acetyl-L-lysine in EHD2 and amphiphysin was performed as described[80,81]. Briefly, 10 mM *N*-(ε)-acetyl-L-lysine and 20 mM nicotinamide were added to the *E. coli* BL21 (DE3) Rosetta culture 30 min before induction of protein expression with IPTG. EHD2 (a.a 1–543, K324amber, see below) and amphiphysin (a.a 1–244, K15amber, see below) were expressed using a pRSFDuet-1 vector co-expressing the tRNA$_{CUA}$ and the acetyl-lysyl-tRNA-synthetase. Mass spectroscopy was used to confirm acetylation of the proteins.

Lysine to glutamine, arginine or amber mutations were made following Quikchange (Agilent) site-directed mutagenesis manufacturer protocols. In order to allow specific spin-labeling of sites on amphiphysin, EHD2 and synaptotagmin1 for EPR experiments, native cysteine residues were mutated to alanines or serines[23,43,67], to create cys-less versions of each protein and site-specific mutation of cysteines was performed in locations known to embed in membranes. In the case of amphiphysin, position 20; for EHD2, position 321; and for synaptotagmin1, position 227 was chosen. The list of primers used in this manuscript is included as Supplementary Table 2, in the Supplementary Information file. Spin label was incubated in a 5- to 10-fold molar excess of protein immediately following the removal of DTT using size exclusion chromatography (PD-10 column (GE)) and left to react at 4°C overnight. Excess spin label was removed using PD-10 columns.

*Vesicle preparation, tubulation assays and electron microscopy.* The initial preparation of vesicles was the same for all lipid compositions used in this study. Lipid stocks were suspended in chloroform and mixed to the desired molar or weight proportions in organic solvent, dried under a stream of $N_2$ gas, and dried overnight in a desiccator. For tubulation TEM studies with amphiphysin and EHD2, multilamellar vesicles (MLVs) of 2:1 wt/wt POPS and POPC, or Folch liposomes were resuspended in buffer A to 4 mg/mL. Protein and lipid were mixed at a 1:375 protein:lipid molar ratio. For EHD2-WT, EHD2-acK324 and EHD2-K324Q, Folch liposomes at a concentration of 1 mg/ml were incubated for 20 min at room temperature with 10 or 7.5 μM protein in the presence of 1 mM ATP. Carbon-coated formvar films mounted on copper grids (Electron Microscopy Services, Hatfield) were suspended on small aliquots of samples for 10 minutes and 30 seconds for amphiphysin and EHD2 studies, respectively. Excess liquid was removed using filter paper and grids were subsequently subjected to a two-minute incubation on a droplet of 1% uranyl acetate was used to stain the sample-coated grids. A JEOL 1400 transmission electron microscope or a Talos L120C was used for specimen observation at 100 kV and 120 kV, respectively. To quantify tubule length, fifteen representative tubule segments were measured and all data was plotted on the scatter plot. Only tubule segments longer than 200 nm were considered for the quantification.

*Liposome co-sedimentation assay.* Sedimentation assays were performed as before[67]. Folch liposomes at a concentration of 2 mg/ml were incubated at room temperature with 10 μM of EHD2-WT, EHD2-acK324, or the K324Q mutant for 20 min in 50 μl reaction volume, followed by a 213,000 g spin for 20 min at 20 °C. The final reaction buffer contained 25 mM HEPES/NaOH (pH 7.5), 300 mM NaCl, 0.5 mM $MgCl_2$ and 1 mM DTT. The supernatant and pellet were subjected to SDS-PAGE. See Source Data for uncropped gels/blots from this and other experiments. The results from three independent experiments were quantified by integrating the protein bands using ImageJ[114] and the intensity of each band (supernatant or pellet) was divided by the sum of the intensities from supernatant and pellet.

*ATPase assays.* ATPase assays were performed as described before[65]. ATPase activities of 10 μM of EHD2-WT, EHD2-acK324 and of the EHD2-K324Q mutant were determined at 30 °C in 25 mM HEPES/NaOH (pH 7.5), 300 mM NaCl, 0.5 mM $MgCl_2$ and 1 mM DTT in the absence and presence of non-extruded Folch liposomes (1 mg/ml final concentration), using 100 μM ATP as the substrate. Reactions were initiated by addition of protein to the reaction. At different time points, reaction aliquots were 5-fold diluted in reaction buffer and quickly transferred to liquid nitrogen. Nucleotides in the samples were separated via a reversed-phase Hypersil ODS-2 C18 column (250 × 4 mm) with HPLC Buffer. Denatured proteins were adsorbed on a C18 guard column. Nucleotides were detected by absorption at 254 nm and quantified by integration of the corresponding peaks. The plots show the average values from three independent experiments, where the error bars correspond to the standard error of the mean.

*Acquisition and analysis of EPR data in lipid binding assays.* Continuous wave (CW) EPR spectra were recorded for samples placed into Quartz capillaries (VitroComInc., New Jersey) using a Bruker EMX spectrophotometer fitted with an ER4119HS resonator. For lipid titration experiments of EHD2, and Syt1 MLVs composed of 3:1 wt/wt POPS:POPC were suspended in 20 mM Hepes, pH 7.4, 150 mM NaCl, as well as 1 mM $Ca^{2+}$ in the case of synaptotagmin1, or 2:1 wt/wt POPG:POPE in the same buffer system for amphiphysin. CW EPR spectral amplitudes were recorded for samples of spin labeled protein in 20 mM Hepes, pH 7.4, 150 mM NaCl buffer. For all experiments, protein concentration was 10 μM and the amount of lipids added was varied. The values were then normalized relative to the protein's CW EPR spectral amplitude from the protein alone in solution. For signal to noise reasons, we recorded the ratio of the amplitude of the immobilized component in the low field transition line and that of the central line width to plot the effect of increasing lipid concentrations.

*Cell culture, transfection and confocal microscopy.* For expression of mutant and wild type amphiphysin in eukaryotic systems, the N-BAR domain was cloned into a pEGFP-N1 vector using 5′ NheI and 3′ XhoI cut sites. For expression of mutant and wild type EHD2 in eukaryotic systems an N-terminally GFP-tagged construct was used, as before[67].

HeLa and COS-7 cell lines were cultured in Dulbecco's Modified Eagle's Medium (DMEM) supplemented with 100 U/mL penicillin G, 100 μg/mL streptomycin, 4.5 g/L glucose, sodium pyruvate (Cellgro, Manassas, VA) and 10% heat inactivated fetal bovine serum (Invitrogen) at 37 °C in humidified at with 5% $CO_2$.

Following three washes with phosphate buffered saline solution, HeLa or COS-7 cells lines were trypsin digested and the cell suspensions were centrifuged at 1000 × g for 5 min. Cells were recovered in fresh media and plated on custom, #1 thickness, glass-bottomed coverslips and allowed to recover for 24 h before transfection.

Expression of cDNA constructs was induced using Lipofectamine 2000 (Invitrogen) and 1.2 μg of cDNA plasmid according to manufacturer protocol. Cells were imaged live at 24 hours following transfection with an Olympus IX-83 confocal microscope using an UPLFN 100x oil immersion objective (NA: 1.30). eGFP fluorescence was excited using a 488 nm laser and light was collected through the objective.

Images were acquired for analysis, which was performed using ImageJ software from the NIH (version 1.48)[114].

*Transgenic fly generation and analysis.* For expression of amphiphysin in flies, full length *Drosophila* amphiphysin was cloned into a pINDY6 vector using XhoI and SpeI. Two constructs were generated, one for wild-type and the other for the acetylation mimicking 5Q15Q mutant. Transgenic flies were generated by standard transformation method[115].

To detect Amph levels in flies, protein extracts were obtained by homogenizing flies in RIPA lysis buffer (50 mM Tris-HCl, pH7.5, 1% NP-40, 0.5% NaDoc, 150 mM NaCl, 0.1% SDS, 2 mM EDTA, 50 mM NaF, 1 mM $Na_3VO_4$, 250 nM cycloporin A, protease inhibitor cocktail (Roche) and phosphatase inhibitor cocktail 1 (Sigma) using mortar and pestle. 15 μg protein homogenate was separated by SDS-PAGE and transferred to nitrocellulose membrane. Primary antibodies were diluted in blocking solution as following: rabbit anti-Amph SH3 domain (from Dr. Harvey McMahon) 1:15000; rabbit anti-Amph 9907 (from Dr. Zelhof) 1:2500; anti-tubulin 1:500 (E7, DSHB).

For immunocytochemistry, indirect flight muscles, IFM, were dissected in PBS and fixed in 4% paraformaldehyde for 25 min. Fixed samples were washed with 0.1% triton X-100 in PBS (PBST) then blocked with 5% normal goat serum (NGS) in PBST. Mouse anti-DLG 1:100 (4F3, DSHB) was diluted in blocking buffer. Alexa-conjugated secondary antibody was used at 1:250 (Invitrogen). 100 nM working stock of fluorescent actin-stain phalloidin (Cytoskeleton) was used to stain actin filaments. Images were captured using Zeiss LSM5 confocal microscope using a 63×1.6NA oil immersion objective with a 2x zoom. When comparing intensity across genotypes, the exposure time was kept constant for all genotypes per experiment.

An adapted "cylinder drop" flight assay was performed similar to that described by Banerjee et al.[116]. A transparent flight chamber with 8-cm inner diameter was made from a plastic transparent sheet. A wider tube that could hold the narrower "drop tube" was attached to the funnel and was placed on top of the flight chamber. 2–4 day old flies were transferred to narrow plastic "drop tube" and 10 to 20 flies per vial were used per drop experiment. Assembled flight chamber was placed on ice to count the number of flies that dropped directly down the chamber, which were considered as defective in flight. The percentage of flight defective flies was calculated by counting the number of flies that dropped to the bottom, divided by the total number of flies in the testing chamber, and then multiplied by 100. At least 5 independent assays were performed per genotype. A two-sided student T-test with assumption of equal variance was used with a cutoff of <0.05% to test statistical significance.

**Reporting summary.** Further information on research design is available in the Nature Research Reporting Summary linked to this article.

## Data availability
All data supporting the findings of this study are available within the article and its supplementary information files. Source data are provided with this paper.

## Code availability
Code used in the bioinformatics studies is available as the Supplementary Data 1 ZIP file.

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

## Acknowledgements

This work was supported by funding from the NIH (GM115736 and NS084345) and the German Research Foundation (SFB958, project A12 and LA 2984/5-1).

## Author contributions

M.R.A., A.K.O., M.L., O.D., K.C., I.H. and R.L. contributed to the conception and design of this study. M.R.A., A.K.O., K.T., J.M.I., J.L., A.M., E.V.S., P.P., D.M., L.B., K.C., O.D. and I.H., performed experiments. A.K.O., M.R.A., K.T., J.L., I.H. and R.L. wrote the manuscript. A.K.O., M.R.A., K.T., J.L., A.M., E.V.S., K.C., O.D., I.H. and R.L. edited the manuscript.

## Competing interests

The authors declare no competing interests.

**Additional information**

