## [Peer Review File · Nature Communications]

Lysine Acetylation Regulates the Interaction between Proteins and MembranesReviewers' comments:

Reviewer #1 (Remarks to the Author):

Okada, et al. provide data from bioinformatic analysis along with biophysical and mutational experiments supporting their postulation that lysine acetylation functions broadly to regulate lipid interactions of peripheral membrane proteins. This is a novel and innovative claim, and their findings could open up a new area of investigation for these important protein families. The bioinformatics data show a clear correlation between previously identified sites of lysine acetylation and known sites of membrane binding in the four protein domain families studied, as mapped onto representative structures in Figure 2. Of course, correlation does not imply causation, and the manuscript does not provide direct evidence that these sites are actively acetylated and deacetylated in cells in order to regulate the strength of membrane binding. Instead, the authors use mutational analysis to demonstrate that neutralization of lysine residues (by mutating to Gln as a mimic of acetylation) weakens lipid interactions and interferes with function *in vivo* and in cultured cells. These results are perhaps not surprising given the largely electrostatic character of these domains' lipid interactions, but strong effects are seen even from single and double mutations. Overall, the lack of direct evidence for a regulatory function related to lipid interaction weakens the conclusions somewhat. However, the circumstantial evidence provided is strong and likely to pave the way for a wide range of future studies in this field.

Concerns:

1. Figure 1 should be revised to reflect normalization based on the number of lysine residues, and the %KAC/K data from Supplementary Table 1 should be moved to the main manuscript, perhaps as a part of Table 1. Because these protein domains interact electrostatically with anionic lipids, the membrane interface regions are enriched in lysines. Therefore, quantifying the frequency of lysine acetylation relative to the total number of amino acids (as done in Figure 1 and Figures S1-S4) overstates the significance of lysine acetylation in the MIR vs. NBR regions. The normalization based on the number of lysine residues (i.e. %KAC/K) is more appropriate.
2. Although the authors generally do a good job of placing their findings in the proper context, there are a few places where the text overstates the evidence. A few examples: (a) the final sentence of the introduction overstates the conclusion that acetylation is used as a regulator of membrane affinity (no direct evidence is provided that acetylation/deacetylation occurs in response to conditions warranting changes in lipid affinity). (b) middle of p. 10, the phrase "acetylation disrupts membrane binding" should be changed to reflect the fact that observations are based on acetylation-mimicking mutations rather than acetylation *per se*.
3. The main text (perhaps the Figure 1 legend) should state how many proteins of each family were used in the bioinformatics analysis. According to the supplementary excel spreadsheet, the analysis included 12 BAR domains, 12 PX domains, 22 C2 domains, and 4 EHD domains.
4. The Discussion overlooks a key peripheral membrane protein whose lipid interactions are known to be affected by lysine acetylation: the tumor suppressor lipid phosphatase PTEN (see, e.g., Okumura et al. JBC 2006 281, 26562-26568). Some reference to this protein is warranted.

Reviewer #2 (Remarks to the Author):

This paper addresses an interesting question that so far has not been studied: how lysine acetylation affects the interaction of peripheral membrane proteins with eukaryotic membranes. Lysine acetylation is a widely-occurring post-translational modification that, with the exception of histones and

some other select proteins, has been under-studied. The authors take advantage of the lysine acetylation data collected on the website Phosphosite.org (mostly this data comes from high-throughput studies). They select 50 proteins from 4 structural families and use existing structural information and sequence comparison to show that lysine acetylation is increased in membrane interacting regions of these proteins. As acetylation significantly changes the chemistry of lysine, the authors propose that this could be a way to directly modulate the interaction of proteins with membranes (as has been demonstrated for phosphorylation). They then perform experiments with four of these proteins to evaluate how lysine modification could affect their membrane-binding properties.

Whereas this first part of the study is very carefully done, I have major reservations regarding the experimental part of this work. This part of the study should be improved.

I. Major concern:

The authors choose four specific examples to test their hypothesis that lysine modification in membrane interacting regions (MIRs) of peripheral memb. proteins affects membrane association. All four examples represent proteins that interact with negatively charged membranes, which are the focus of this study. In all four examples, the authors use glutamine as a mimic of acetyl-lysine. They show that replacing select lysine residues, which have been reported to be acetylated, with glutamine abrogates membrane interaction.

I agree that, among the 20 universal amino acids, Lys to Gln substitution is the best mimic for Lys acetylation. But Gln is actually structurally quite different from acetyl-Lys (shorter by 3 carbon groups, making it considerably less hydrophobic!). I therefore wonder to what extent Gln really is a good mimic of acetyl-Lys in the case of protein-membrane interactions.

In this respect, it is of note that already over a decade ago, the S. Munro and C. Burd labs observed that N-acetylation of Golgi-acting proteins Arl3 and Grh1 was required for their Golgi association; in their un-acetylated form, the proteins were cytosolic. So, while the experiments in this manuscript might suggest that Lys acetylation prevents membrane binding, the opposite was observed in the case of these N-acetylated peripheral membrane proteins. N-acetylation was also shown to increase the folding of the α -synuclein helix, which should increase its membrane binding. With these examples and the structure of acetyl-Lys in mind, I am left wondering what the effect of Lys-acetylation would actually be on the behavior of the proteins that the authors study: decrease in membrane-binding, change in their membrane specificity, change in dynamics of membrane association?

Of course, a single manuscript cannot solve all questions, but the authors' reliance on Gln to Lys substitution is over-simplistic and may be detrimental to further progress in the field. For the experiments that are presented in the manuscript now, my interpretation would be simply that decreasing the positive charge of these proteins via Lys to Gln substitutions decreases their interaction with negatively-charged membranes; the importance of electrostatic interactions for membrane binding is amply demonstrated in the literature, and therefore these observations don't seem very novel.

Two types of experiments would strengthen this study:

1. Some evaluation of the effect of Lys-acetylation per se on protein-membrane interaction in vitro. I realize that this is much less straightforward than using genetically encoded glutamine. If an in vitro experiment is deemed too difficult, molecular dynamics simulations could also be used here.

2. Some demonstration of changes in the level of acetylation of one of the proteins that the authors study in a cellular context should be added, ideally linked to a change in cellular phenotype. (This type of experiment could also be a way to evaluate how well Gln mimics acetyl-Lys.) It is plausible that reversible lysine acetylation would be a way to regulate membrane association of these proteins, but currently the authors do not present sufficient evidence for this statement.

II. Other comments:

1. Supplementary Table 1 shows that not only is Lys acetylation increased in MIRs of the studied proteins, but also percent of Lys is higher than in NBRs. This is important, so I think that the information on the total lysine content should be included in the main figures. I also wonder here if it is the Lys content that is increased in these MIRs, or positive charge, so a comparison between Lys and Arg content would be interesting. Lys has been suggested to have other benefits for membrane binding compared to Arg, not just the possibility of regulation through acetylation suggested here; nevertheless, comparison between Lys and Arg content seems relevant for the authors' hypothesis.

2. In Fig. 3, two liposome compositions are used to evaluate amphiphysin-membrane interaction, giving qualitatively the same results. Since POPG:POPE liposomes are not physiologically relevant, I don't see the point of including experiments with these liposomes. Modulating physiologically relevant membrane properties in this experiment would be interesting, but it is more important to first establish how informative are the K-Q substitutions.

3. In Suppl Fig 4e, DYSF and ESYT1 appear twice; it should be indicated which part of the protein each entry refers to. Also, panel 4f has no figure legend.

Reviewer #3 (Remarks to the Author):

The inner leaflet of the plasma membrane (PM) is negatively charged as a consequence of both phosphorylation of phosphoinositides and, more importantly, the asymmetry of phosphatidylserine across the bilayer. Not surprisingly, this charge distribution is exploited as an important mechanism whereby peripheral membrane proteins associate with the PM through electrostatic interactions by displaying a positive charge in their membrane association domains. The advantage of this mechanism is that it can be easily regulated by modulating the charge of the inner leaflet (e.g. by PI3K or PTEN) or by partially neutralizing the basic domains of peripheral membrane proteins. One clear example (not cited) is the reversal of membrane association of KRAS4B, driven by its C-terminal polylysine motif, by phosphorylation of serine 181. Among the ways that the positive charge of peptides containing lysine can be modulated is through lysine acetylation. With advances in proteomics over the past 15 years lysine acetylation has been recognized as an exceedingly common post-translational modification of cellular proteins. Accordingly, one would predict that lysine acetylation can modulate membrane association of a subset of peripheral membrane proteins. This is the hypothesis tested by the authors who examined four families of peripheral membrane proteins in which the membrane-association domain is well defined. The authors mined the Phosphosite.org database of protein lysine acetylation data gathered largely through published proteomic analysis. They found that lysine acetylation was much more common in the membrane association domain than elsewhere in the polypeptide, suggesting regulation by this post-translational modification. They went on to test this hypothesis by changing lysines to glutamines in the membrane association domains of a number of peripheral membrane proteins and assaying for membrane association with a variety of methods including the induction of membrane curvature and membrane association of GFP-tagged proteins. Indeed, charge reduction had a profound effect.

Although the idea of lysine acetylation modulating membrane association of peripheral membrane proteins is rather obvious, the strength of the current study is the explication of and exploration of this cell biological truism. The analysis of available public domain proteomic data is interesting and well done, although it cannot rise above the level of association to the level of causality. The weakness of the study is the simple substitution of glutamine for lysine to explore the role of lysine acetylation. There is no doubt that the positive charge of lysine side chains contributes to the association of a large subset of peripheral membrane proteins with the inner leaflet of the PM and that the basis for this is charge. Thus, negating the charge in any manner, such as substitution of glutamine (but not arginine)

for lysine, would be expected to have an effect. Indeed, this is confirmed in a compelling way by the data presented. However, this result does not establish acetylation as the mechanism of charge modulation *in vivo*. I do not mean to say that acetylation is not the mechanism of physiologic charge neutralization. It is by far the most likely cause. But the experimental results do not prove the point. They speak only to charge and are silent on acetylation. That said, a direct demonstration of acetylation as the direct cause of charge neutralization and resulting loss of affinity for membranes is a tall order. Since one cannot monitor lysine acetylation in real time in living cells while monitoring the distribution of proteins of interest, perhaps the best bet would be to look for the effect of regulators of acetylation on membrane association of target proteins. If the authors could determine what are the relevant acetyltransferases or deacetylases for a given membrane association domain such as the BAR domains studied, then monitoring membrane association upon up or down regulation of these regulatory enzymes would be highly informative. Alas, since neither the relevant acetyltransferase nor deacetylase are known, this is not a line of investigation that can be employed.

Reviewer #4 (Remarks to the Author):

This manuscript uses various approaches to assess the potential role of lysine acetylation as regulatory mechanism for protein function. The authors present several new lines of showing that acetylation mainly affects protein binding sites to lipid membranes. They establish that mutations mimicking acetylation in amphiphysin, Syt1 and EHD2 strongly reduced membrane binding affinity, and associated membrane remodeling of both amphiphysin and EHD2. Further, they found that mimicking a single acetylation site of amphiphysin and EHD2 reduce binding affinity and promote loss of plasma membrane targeting. They used *Drosophila* and showed that expression of Amph-5Q15Q caused major disruption of the T-tubule network with a clear loss of flight.

The fact that charge neutralization of key lysine residues is key to control membrane interaction is well supported by well conducted experiments.

Major issues

I find this study interesting but limited in scope as it does not address the critical point of whether acetylation is functionally relevant. In other words, in which physiological or pathological conditions do acetylation promote a loss of membrane binding? Can the authors provide a demonstration that upon acetylation (not just mimicking it by mutation) the studied domains are exiting membranes? They should also show that blocking acetylation prevents these domains from exiting the membrane in relevant physiological conditions (that promote acetylation). This is a major issue that I feel the authors should resolve prior publication in *Nature Communications*.

Minor issues:

- Fig 1: Enrichment acetylation sites within MIRs: this figure is interesting but I wondered whether it would be possible to normalize to the actual number of lysines present in the protein. This would allow to have a true representation of the enrichment per protein.
- How effective is the K5Q-K15Q mutation in mimicking acetylation in *Drosophila* amphiphysin (Amph-5Q15Q)? This requires additional controls.
- Many studies have assessed the role of lysine in synaptotagmin C2 domains and several seminal papers should be cited.

Reviewer #1 (Remarks to the Author):

Okada, et al. provide data from bioinformatic analysis along with biophysical and mutational experiments supporting their postulation that lysine acetylation functions broadly to regulate lipid interactions of peripheral membrane proteins. This is a novel and innovative claim, and their findings could open up a new area of investigation for these important protein families. The bioinformatics data show a clear correlation between previously identified sites of lysine acetylation and known sites of membrane binding in the four protein domain families studied, as mapped onto representative structures in Figure 2. Of course, correlation does not imply causation, and the manuscript does not provide direct evidence that these sites are actively acetylated and deacetylated in cells in order to regulate the strength of membrane binding. Instead, the authors use mutational analysis to demonstrate that neutralization of lysine residues (by mutating to Gln as a mimic of acetylation) weakens lipid interactions and interferes with function in vivo and in cultured cells. These results are perhaps not surprising given the largely electrostatic character of these domains' lipid interactions, but strong effects are seen even from single and double mutations. Overall, the lack of direct evidence for a regulatory function related to lipid interaction weakens the conclusions somewhat. However, the circumstantial evidence provided is strong and likely to pave the way for a wide range of future studies in this field.

We thank the reviewer for recognizing the novel findings from our study that are likely to open new areas of investigation into the function and regulation of these protein families.

Concerns:

1. Figure 1 should be revised to reflect normalization based on the number of lysine residues, and the %KAC/K data from Supplementary Table 1 should be moved to the main manuscript, perhaps as a part of Table 1. Because these protein domains interact electrostatically with anionic lipids, the membrane interface regions are enriched in lysines. Therefore, quantifying the frequency of lysine acetylation relative to the total number of amino acids (as done in Figure 1 and Figures S1-S4) overstates the significance of lysine acetylation in the MIR vs. NBR regions. The normalization based on the number of lysine residues (i.e. %KAC/K) is more appropriate.

We have included these data in the main manuscript as part of Table 1.

2. Although the authors generally do a good job of placing their findings in the proper context, there are a few places where the text overstates the evidence. A few examples: (a) the final sentence of the introduction overstates the conclusion that acetylation is used as a regulator of membrane affinity (no direct evidence is provided that acetylation/deacetylation occurs in response to conditions warranting changes in lipid affinity). (b) middle of p. 10, the phrase "acetylation disrupts membrane binding" should be changed to reflect the fact that observations are based on acetylation-mimicking mutations rather than acetylation per se.

We thank the reviewer for this point and agree that the statements in the original manuscript benefits from additional support. We have now developed acetylated forms of EHD2 and amphiphysin to directly test this point. Indeed, we find that acetylation reduces membrane binding and function of both proteins. In conjunction with our bioinformatics data which argue strongly that acetylation is much more prevalent in membrane binding regions, these data

support a regulatory role of acetylation on membrane-protein function. In addition, we have softened the language in the indicated sentence of the introduction to say, “Taken together, our data suggest that acetylation can regulate protein-membrane interactions and membrane protein function”, while the wording “acetylation disrupts membrane binding” is now substantiated by the new studies.

3. The main text (perhaps the Figure 1 legend) should *state how many proteins of each family were used in the bioinformatics analysis.* According to the supplementary excel spreadsheet, the analysis included 12 BAR domains, 12 PX domains, 22 C2 domains, and 4 EHD domains.

As suggested by the reviewer, we have added a line to the legend of figure 1 with this information.

4. The Discussion overlooks a key peripheral membrane protein whose lipid interactions are known to be affected by lysine acetylation: the tumor suppressor lipid phosphatase PTEN (see, e.g., Okumura et al. JBC 2006 281, 26562-26568). Some reference to this protein is warranted.

We agree with the reviewer that the study from Okumura et al. is quite pertinent to our findings and appreciate the insight. We have now included a small section in our discussion regarding PTEN in paragraph 2, sentences 15 and 16 as follows. “For instance, in proteins interacting with highly negatively charged lipids (e.g. PI with multiple phosphates), the loss of positive charge via acetylation may shift the specificity to a less phosphorylated PI or even neutral lipid. While this would be expected to have effects on membrane localization, it may also affect enzymatic activity. In fact, the loss of ATPase activity we see with acK324-EHD2 in our studies presented here is one such example. Similar mechanisms might also apply to PTEN, which is acetylated in its catalytic phosphatase domain at K163/K164 where acetylation attenuates enzymatic activity.”

Reviewer #2 (Remarks to the Author):

This paper addresses an interesting question that so far has not been studied: how lysine acetylation affects the interaction of peripheral membrane proteins with eukaryotic membranes. Lysine acetylation is a widely-occurring post-translational modification that, with the exception of histones and some other select proteins, has been under-studied. The authors take advantage of the lysine acetylation data collected on the website Phosphosite.org (mostly this data comes from high-throughput studies). They select 50 proteins from 4 structural families and use existing structural information and sequence comparison to show that lysine acetylation is increased in membrane interacting regions of these proteins. As acetylation significantly changes the chemistry of lysine, the authors propose that this could be a way to directly modulate the interaction of proteins with membranes (as has been demonstrated for phosphorylation). They then perform experiments with four of these proteins to evaluate how lysine modification could affect their membrane-binding properties.

Whereas this first part of the study is very carefully done, I have major reservations regarding the experimental part of this work. This part of the study should be improved.

I. Major concern:

The authors choose four specific examples to test their hypothesis that lysine modification in membrane interacting regions (MIRs) of peripheral memb. proteins affects membrane association. All four examples represent proteins that interact with negatively charged membranes, which are the focus of this study. In all four examples, the authors use glutamine as a mimic of acetyl-lysine. They show that replacing select lysine residues, which have been reported to be acetylated, with glutamine abrogates membrane interaction.

I agree that, among the 20 universal amino acids, Lys to Gln substitution is the best mimic for Lys acetylation. But Gln is actually structurally quite different from acetyl-Lys (shorter by 3 carbon groups, making it considerably less hydrophobic!). I therefore wonder to what extent Gln really is a good mimic of acetyl-Lys in the case of protein-membrane interactions.

In this respect, it is of note that already over a decade ago, the S. Munro and C. Burd labs observed that N-acetylation of Golgi-acting proteins Arl3 and Grh1 was required for their Golgi association; in their un-acetylated form, the proteins were cytosolic. So, while the experiments in this manuscript might suggest that Lys acetylation prevents membrane binding, the opposite was observed in the case of these N-acetylated peripheral membrane proteins. N-acetylation was also shown to increase the folding of the α -synuclein helix, which should increase its membrane binding. With these examples and the structure of acetyl-Lys in mind, I am left wondering what the effect of Lys-acetylation would actually be on the behavior of the proteins that the authors study: decrease in membrane-binding, change in their membrane specificity, change in dynamics of membrane association?

The main point brought up here, that it is critical to directly test the effects of lysine acetylation on the behavior of the proteins we are studying, was noted among all of the reviewers. We have now developed acetylated proteins and directly tested the effects and find the results similar in nature to our studies with acetylation mimetics. These data are shown in Figures 4 and 6.

Of course, a single manuscript cannot solve all questions, but the authors' reliance on Gln to Lys substitution is over-simplistic and may be detrimental to further progress in the field. For the experiments that are presented in the manuscript now, my interpretation would be simply that decreasing the positive charge of these proteins via Lys to Gln substitutions decreases their interaction with negatively-charged membranes; the importance of electrostatic interactions for membrane binding is amply demonstrated in the literature, and therefore these observations don't seem very novel.

Two types of experiments would strengthen this study:

1. Some evaluation of the effect of Lys-acetylation per se on protein-membrane interaction in vitro. I realize that this is much less straightforward than using genetically encoded glutamine. If an in vitro experiment is deemed too difficult, molecular dynamics simulations could also be used here.

We chose to obtain the data directly as discussed above.

2. Some demonstration of changes in the level of acetylation of one of the proteins that the authors study in a cellular context should be added, ideally linked to a change in cellular phenotype. (This type of experiment could also be a way to evaluate how well Gln mimics acetyl-Lys.) It is plausible that reversible lysine acetylation would be a way to regulate membrane association of these proteins, but currently the authors do not present sufficient evidence for this statement.

We agree with the reviewer that it would be nice to have this data. On the other hand, our bioinformatics results already show that acetylation is enhanced in membrane interacting regions. The individual enzymes and pathways that would control and regulate these effects are unknown at this time. We hope that our study will inspire further investigation into this area. However, we agree with reviewer 3 that since neither the relevant acetyltransferases nor deacetylases are known, further investigation into these pathways are beyond the scope of this paper and that a single paper cannot address all questions.

II. Other comments:

1. Supplementary Table 1 shows that not only is Lys acetylation increased in MIRs of the studied proteins, but also percent of Lys is higher than in NBRs. This is important, so I think that the information on the total lysine content should be included in the main figures. I also wonder here if it is the Lys content that is increased in these MIRs, or positive charge, so a comparison between Lys and Arg content would be interesting. Lys has been suggested to have other benefits for membrane binding compared to Arg, not just the possibility of regulation through acetylation suggested here; nevertheless, comparison between Lys and Arg content seems relevant for the authors' hypothesis.

We have included the data regarding acetylation relative to lysine content in the main manuscript as part of Table 1. Regarding the question of arginine and lysine comparison, this is an interesting question and one that we feel should be part of an independent study.

2. In Fig. 3, two liposome compositions are used to evaluate amphiphysin-membrane interaction, giving qualitatively the same results. Since POPG:POPE liposomes are not physiologically relevant, I don't see the point of including experiments with these liposomes. Modulating physiologically relevant membrane properties in this experiment would be interesting, but it is more important to first establish how informative are the K-Q substitutions.

We agree that POPG:POPE is not physiologically relevant for most eukaryotic systems and we now only show the tubulation of the physiologically relevant POPS:POPC data.

3. In Suppl Fig 4e, DYSF and ESYT1 appear twice; it should be indicated which part of the protein each entry refers to. Also, panel 4f has no figure legend.

We agree that the initial labeling scheme was confusing. We have now labelled the apparently duplicated name differently to emphasize the distinction, which now also correlates to the naming in the supplemental spreadsheet 1, where detailed information regarding the definitions of MIRs and NBRs for all domains studied is reported. Furthermore, we have included wording in the supplemental figure 2-5 legends as follows, "See supplemental spreadsheet 1 for details including definitions of MIRs and NBRs for each domain" to help guide the reader. We have updated the missing part of the legend of Supplemental Fig 4f and want to thank the reviewer for catching this.

Reviewer #3 (Remarks to the Author):

The inner leaflet of the plasma membrane (PM) is negatively charged as a consequence of both phosphorylation of phosphoinositides and, more importantly, the asymmetry of phosphatidylserine across the bilayer. Not surprisingly, this charge distribution is exploited as an important mechanism whereby peripheral membrane proteins associate with the PM through electrostatic interactions by displaying a positive charge in their membrane association domains. The advantage of this mechanism is that it can be easily regulated by modulating the charge of the inner leaflet (e.g. by PI3K or PTEN) or by partially neutralizing the basic domains of peripheral membrane proteins. One clear example (not cited) is the reversal of membrane association of KRAS4B, driven by its C-terminal polylysine motif, by phosphorylation of serine 181.

We now include a reference to KRAS4B in the introduction of our manuscript in the 2nd paragraph, 3rd sentence as follows, “Not surprisingly, phosphorylation, the most well-studied of the PTMs, can exert influence over protein-membrane interactions by changing the charge potential of membrane-exposed acidic residues 9–12”. The KRAS4B citation is number 12.

Among the ways that the positive charge of peptides containing lysine can be modulated is through lysine acetylation. With advances in proteomics over the past 15 years lysine acetylation has been recognized as an exceedingly common post-translational modification of cellular proteins. Accordingly, one would predict that lysine acetylation can modulate membrane association of a subset of peripheral membrane proteins. This is the hypothesis tested by the authors who examined four families of peripheral membrane proteins in which the membrane-association domain is well defined. The authors mined the Phosphosite.org database of protein lysine acetylation data gathered largely through published proteomic analysis. They found that lysine acetylation was much more common in the membrane association domain than elsewhere in the polypeptide, suggesting regulation by this post-translational modification. They went on to test this hypothesis by changing lysines to glutamines in the membrane association domains of a number of peripheral membrane proteins and assaying for membrane association with a variety of methods including the induction of membrane curvature and membrane association of GFP-tagged proteins. Indeed, charge reduction had a profound effect.

Although the idea of lysine acetylation modulating membrane association of peripheral membrane proteins is rather obvious, the strength of the current study is the explication of and exploration of this cell biological truism. The analysis of available public domain proteomic data is interesting and well done, although it cannot rise above the level of association to the level of causality. The weakness of the study is the simple substitution of glutamine for lysine to explore the role of lysine acetylation. There is no doubt that the positive charge of lysine side chains contributes to the association of a large subset of peripheral membrane proteins with the inner leaflet of the PM and that the basis for this is charge. Thus, negating the charge in any manner, such as substitution of glutamine (but not arginine) for lysine, would be expected to have an effect. Indeed, this is confirmed in a compelling way by the data presented. However, this result does not establish acetylation as the mechanism of charge modulation in vivo. I do not mean to say that acetylation is not the mechanism of physiologic charge neutralization. It is by far the most likely cause. But the experimental results do not prove the point. They speak only to charge and are silent on acetylation. That said, a direct demonstration of acetylation as the direct cause of charge neutralization and resulting loss of affinity for membranes is a tall order. Since one cannot monitor lysine acetylation in real time in living cells while monitoring the distribution of proteins of interest, perhaps the best bet would be to look for the effect of regulators of acetylation on membrane association of target proteins. If the authors could

determine what are the relevant acetyltransferases or deacetylases for a given membrane association domain such as the BAR domains studied, then monitoring membrane association upon up or down regulation of these regulatory enzymes would be highly informative. Alas, since neither the relevant acetyltransferase nor deacetylase are known, this is not a line of investigation that can be employed.

We certainly appreciate reviewer 3's acknowledgement of the difficulties of directly demonstrating acetylation as a cause of changes in membrane affinities. In fact, it did take us some time to develop the tools to test this explicitly, but with the addition of our new data in Figs. 4 & 6, we feel our data now demonstrates that acetylation does indeed directly change membrane-binding affinity and membrane remodeling.

As discussed above in response to reviewer 2, we agree it would be very informative to monitor membrane association of BAR or other domains upon up or down regulation of relevant acetyltransferases or deacetylases. We concur with reviewer 3 that "since neither the relevant acetyltransferase nor deacetylase are known, this is not a line of investigation that can be employed." We feel that identification and characterization of these enzymes will be an exciting new field of study that is predicted by this work. The final paragraph of the conclusion emphasizes that this is an important next step following our initial study.

Reviewer #4 (Remarks to the Author):

This manuscript uses various approaches to assess the potential role of lysine acetylation as regulatory mechanism for protein function. The authors present several new lines of showing that acetylation mainly affects protein binding sites to lipid membranes. They establish that mutations mimicking acetylation in amphiphysin, Syt1 and EHD2 strongly reduced membrane binding affinity, and associated membrane remodeling of both amphiphysin and EHD2. Further, they found that mimicking a single acetylation site of amphiphysin and EHD2 reduce binding affinity and promote loss of plasma membrane targeting. They used Drosophila and showed that expression of Amph-5Q15Q caused major disruption of the T-tubule network with a clear loss of flight.

The fact that charge neutralization of key lysine residues is key to control membrane interaction is well supported by well conducted experiments.

Major issues

I find this study interesting but limited in scope as it does not address the critical point of whether acetylation is functionally relevant. In other words, in which physiological or pathological conditions do acetylation promote a loss of membrane binding? Can the authors provide a demonstration that upon acetylation (not just mimicking it by mutation) the studied domains are exiting membranes? They should also show that blocking acetylation prevents these domains from exiting the membrane in relevant physiological conditions (that promote acetylation). This is a major issue that I feel the authors should resolve prior publication in Nature Communications.

We have now added data that directly demonstrate EHD2 exiting membranes when acetylated in in vitro conditions with physiologically relevant membranes. With respect to doing this in vivo, we agree with reviewer 3 that one would have to know the relevant acetyltransferases and deacetylases involved in these pathways and we feel that identifying these pathways is beyond the scope of this study.

Minor issues:

- Fig 1: Enrichment acetylation sites within MIRs: this figure is interesting but I wondered whether it would be possible to normalize to the actual number of lysines present in the protein. This would allow to have a true representation of the enrichment per protein.

We have modified Table 1 to include this data and the enrichment persists when normalized to the number of lysines.

- How effective is the K5Q-K15Q mutation in mimicking acetylation in Drosophila amphiphysin (Amph-5Q15Q)? This requires additional controls.

We have now verified that acetylated amphiphysin inhibits tubulation just as the Amph-5Q15Q acetylation mimic and have included this data in Fig. 6.

- Many studies have assessed the role of lysine in synaptotagmin C2 domains and several seminal papers should be cited.

We thank the reviewer for pointing this out and have included several references that can be found in the reference section as follows; references 60 – 66.

REVIEWER COMMENTS

Reviewer #2 (Remarks to the Author):

The authors have done thorough and honest revision of their manuscript and they have addressed all concerns of the reviewers. In particular, they added experiments using acetylated purified EHD2 and amphiphysin, which complement their previous work. I congratulate the authors on this important and inspiring work, which will undoubtedly form the basis for many future studies.

I have only one larger comment:

Not surprisingly, the data presented in Figs. 3 & 4 show that the K324Q mutant of EHD2 does not behave in the same way as EHD2-ack324, suggesting limitations to using Q as a mimic of ac-K. I would appreciate more discussion of this point.

Differences in the behavior of EHD2-ack324 and K324Q do not in any way diminish the importance and elegance of this study; on the contrary. The limitations of the Q mimic should be discussed more openly.

Some quantification of the differences between EHD2-WT and EHD2-ack324 in liposomes tubulation would be useful.

Finally, I am surprised by the large difference in the tubulation activity of EHD2-K324Q and EHD2-ack324, given that they show a similar amount of liposome binding. Can this be explained by the factor of stimulation of the ATPase activity by membranes for the two mutants. In the liposome tubulation assay, doesn't the ack324 mutant appear more active than EHD2-WT? How can this data be reconciled with the lower liposome binding and lower ATPase activity of ack324 compared to EHD2-WT?

Minor points:

Table 1: last column is messed up.

Figure 1:

- Y-axis label: percentage of what?

Figure 3:

- B: Please make the data points stand out more (+ use color for consistency with panel C). I don't see the benefit of showing the connecting lines between individual data points; I would suggest removing them.

- Data points are too small and difficult to separate; please use color (consistent with panel B)

Some mistakes in the references; Refs. 14 and 15 are listed twice.

Reviewer #3 (Remarks to the Author):

NCOMMS-18-07268A

Okada et al. have revised their manuscript that reports modulation of membrane interaction regions (MIRs) of peripheral membrane proteins by lysine acetylation. All four reviewers found the same deficit in the work. Although the bioinformatic analysis was interesting, no direct demonstration of lysine acetylation affecting membrane association was provided. Instead the authors relied on substitution of glutamine for lysine, which is considered an acetylation mimic. However, because membrane association of this class of protein is driven by the electrostatic interaction between polybasic patches of the relevant protein that are created in part by lysines, loss of the charge by glutamine substitution implicated charge but was silent with regard to acetylation. The authors have

addressed this criticism with new experiments in which they generate EHD2 and amphiphysin in *E. coli* with site-specific incorporation of N-(ϵ)-acetyl-L-lysine. They apply these recombinant proteins, along with WT and K>Q mutants, to liposome preparations and observed tubulation by TEM. This is a reasonable way to address the criticism, however the results are not compelling. Whereas acetylated amphiphysin was indeed as deficient as was K>Q amphiphysin in tubulating membranes (Fig. 6), acetylated EH2 showed robust tubulation in stark contrast to K>Q EH2 (Fig. 4). The authors interpret the appearance of small, ring-like structures around the tubulated vesicles in the condition in which EHD2-acK324 was used as indicating an "altered binding mode." I am not sure what they mean by this but what is clear is that EHD2-acK324 does not phenocopy EHD2-K324Q in this assay. The similar loss of function of Amph-acK15 and Amph-5Q15Q notwithstanding, the total of the evidence does not support the conclusion.

Not only is a direct demonstration of the effect of lysine acetylation missing but there is also no evidence presented for dynamic acetylation, which would be the hallmark of a regulatory pathway. The authors' hypothesis is that lysine acetylation, a common post-translational modification of proteins, can regulate the MIRs of peripheral membrane proteins that bind to membranes via an electrostatic interaction. Conditions for the translocation to membrane of many such proteins, e.g. Eps15, are well documented. If lysine acetylation modulates this process perhaps the authors could affinity capture an epitope tagged protein of interest from cell lysates with or without stimulating translocation and look by mass spectroscopy for alterations in the acetylation of the MIR of the protein of interest.

Reviewer #4 (Remarks to the Author):

The authors have satisfactorily addressed all my queries.

Reviewer #2 (Remarks to the Author):

The authors have done thorough and honest revision of their manuscript and they have addressed all concerns of the reviewers. In particular, they added experiments using acetylated purified EHD2 and amphiphysin, which complement their previous work. I congratulate the authors on this important and inspiring work, which will undoubtedly form the basis for many future studies.

I have only one larger comment:

Not surprisingly, the data presented in Figs. 3 & 4 show that the K324Q mutant of EHD2 does not behave in the same way as EHD2-acK324, suggesting limitations to using Q as a mimic of ac-K. I would appreciate more discussion of this point.

Differences in the behavior of EHD2-acK324 and K324Q do not in any way diminish the importance and elegance of this study; on the contrary. The limitations of the Q mimic should be discussed more openly. Some quantification of the differences between EHD2-WT and EHD2-acK324 in liposomes tubulation would be useful.

Finally, I am surprised by the large difference in the tubulation activity of EHD2-K324Q and EHD2-acK324, given that they show a similar amount of liposome binding. Can this be explained by the factor of stimulation of the ATPase activity by membranes for the two mutants. In the liposome tubulation assay, doesn't the acK324 mutant appear more active than EHD2-WT? How can this data be reconciled with the lower liposome binding and lower ATPase activity of acK324 compared to EHD2-WT?

*We agree with reviewer 2 that the differences in behavior between the mutant and acetylated form of EHD2 “do not in any way diminish the importance and elegance of this study”. Indeed, it is well-recognized that glutamine mimetics are imperfect, and this is the reasoning behind providing additional data in the last revision. We also agree that the manuscript would benefit from augmentation of the discussion regarding the difference between EHD2-acK324 and EHD2-K324Q in tubulation. We have obtained additional experimental data to more carefully tease out the differences between the three different EHD2 variants at different concentrations and added a section to the discussion (3rd to last paragraph in the discussion) as recommended. These data also include additional quantification as requested. We have added a new Fig. 4e to show the pronounced difference in tubule length under more stringent conditions (lower protein concentrations), suggesting destabilization of tubules by EHD2-acK324. This also helps to explain the vesiculation seen previously, which continues to be observed. The quantification shows a significant difference in membrane remodeling between the EHD2-WT and the acetylated protein and an even stronger effect was observed for the EHD2-K324Q mutant. These data are presented in the results section labeled “**Acetylation and acetylation mimetics of EHD2 decrease membrane affinity, binding, catalytic activity and alter its membrane remodeling**”, fourth paragraph. The data now provide a quantitative measure showing that (a) acetylation strongly affects function and (b) that this effect is even stronger for the EHD2-K324Q mutant.*

Thus, the totality of our data indicates that acetylation as well as gln mimicry both have strong effects on membrane interaction. We note that our studies in figure 3 help to define these differences better by showing both decreased membrane binding affinity and enzymatic activity from both the glutamine mimic and acetylated protein.

In the case of amphiphysin, we find glutamine and acetylated lysine behave essentially the same. This allowed us to utilize our amphiphysin glutamine mimic in cellular and drosophila studies, which is the strength of the mutational approach.

Minor points:

Table 1: last column is messed up.

No specific issues were found with the last column of table 1 in the file, pre-upload. We suspect that a formatting error during upload lead to this and will double check during this submission.

Figure 1:

- Y-axis label: percentage of what?

The Y-axis label in Figure 1 has been changed to say "Percentage of domains"

Figure 3:

- B: Please make the data points stand out more (+ use color for consistency with panel C). I don't see the benefit of showing the connecting lines between individual data points; I would suggest removing them.

- Data points are too small and difficult to separate; please use color (consistent with panel B).

We have increased the size of the icons for each data point and added color, consistent between panels B and C as suggested.

Some mistakes in the references; Refs. 14 and 15 are listed twice.

References 14 and 15 were in fact repeated at references 18 and 87. This was fixed.

Reviewer #3 (Remarks to the Author):

NCOMMS-18-07268A

Okada et al. have revised their manuscript that reports modulation of membrane interaction regions (MIRs) of peripheral membrane proteins by lysine acetylation. All four reviewers found the same deficit in the work. Although the bioinformatic analysis was interesting, no direct demonstration of lysine acetylation affecting membrane association was provided. Instead the authors relied on substitution of glutamine for lysine, which is considered an acetylation mimic. However, because membrane association of this class of protein is driven by the electrostatic interaction between polybasic patches of the relevant protein that are created in part by lysines, loss of the charge by glutamine substitution implicated charge but was silent with regard to acetylation. The authors have addressed this criticism with new experiments in which they generate EHD2 and amphiphysin in *E. coli* with site-specific incorporation of N-(ϵ)-acetyl-L-lysine. They apply these recombinant proteins, along with WT and K>Q mutants, to liposome preparations and observed tubulation by TEM. This is a reasonable way to address the criticism, however the results are not compelling. Whereas acetylated amphiphysin was indeed as deficient as was K>Q amphiphysin in tubulating membranes (Fig. 6), acetylated EH2 showed robust tubulation in stark contrast to K>Q EH2 (Fig. 4). The authors interpret the appearance of small, ring-like structures around the tubulated vesicles in the condition in which EHD2-acK324 was used as indicating an “altered binding mode.” I am not sure what they mean by this but what is clear is that EHD2-acK324 does not phenocopy EHD2-K324Q in this assay. The similar loss of function of Amph-acK15 and Amph-5Q15Q notwithstanding, the total of the evidence does not support the conclusion.

Reviewer 3 points out that we have addressed the previous criticisms in a reasonable way and we appreciate this as great effort went into adding this component to the paper, but reviewer 3 also finds that the lack of phenocopy between the EHD2-acK324 and the glutamine mimic in the tubulation assay diminishes the study. As we discussed in response to reviewer 2, we have performed additional experiments that more quantitatively assess the differences in tubulation between the three EHD2 variants. We now quantitatively address reviewer 3’s concern about altered binding modes by determining the different tubule lengths of EHD2-WT and acetylated protein shown in Figure 4e, demonstrating destabilization of tubules by acetylation. We continue to observe small vesicular structures around the shortened tubules, indicative of increased vesiculation. Analogous observations have previously been reported for a phosphorylation mimicking mutant of endophilin (S75D), and this study as well as three others are now cited. We also note the experiments presented in Figure 3 demonstrate differences between WT and the acetylated and gln variants in binding, affinity and catalytic activity, which all help to support the conclusion that acetylation has an effect on membrane binding properties.

While we find that the membrane binding properties of the acetylated protein and the EHD2-K324Q mutant are strongly affected, the effects are not always identical, as reviewer 3 points out. Therefore, as discussed in our response to reviewer 2, we have included a section in the discussion (3rd to last paragraph in the discussion) directly addressing that gln mutants may not always behave exactly in the same way as the acetylated protein. In cases where acetylation and gln mutations have a comparable effect, it becomes possible to genetically encode the acetylation effects via gln mutations in cell and animal studies.

Here, we feel the overall similarity in the multiple biochemical aspects tested support the validity of using a mimetic in cellular assays with the caveat that mimetics are useful tools with limitations.

Not only is a direct demonstration of the effect of lysine acetylation missing but there is also no evidence presented for dynamic acetylation, which would be the hallmark of a regulatory pathway. The authors' hypothesis is that lysine acetylation, a common post-translational modification of proteins, can regulate the MIRs of peripheral membrane proteins that bind to membranes via an electrostatic interaction. Conditions for the translocation to membrane of many such proteins, e.g. Eps15, are well documented. If lysine acetylation modulates this process perhaps the authors could affinity capture an epitope tagged protein of interest from cell lysates with or without stimulating translocation and look by mass spectroscopy for alterations in the acetylation of the MIR of the protein of interest.

We now provide qualitative and quantitative evidence that acetylation affects protein-membrane interaction. We have qualified our previous statement regarding the charge-neutralization mechanism of acetylation to say that it is an important, but by no means the only factor by which acetylation affects protein-membrane interaction. In the case of amphiphysin, we find that gln modifications faithfully mimic acetylation. This allows us to demonstrate the pronounced effect of acetylation in cells and in vivo. Together with our bioinformatics data, which show a high regulatory activity of acetylation specifically in membrane-interacting regions, we conclude that acetylation is a likely regulator of protein-membrane function by modulating membrane-interaction. We feel this is already an important contribution that will herald many further studies. We also acknowledge that there are many open questions regarding the dynamic regulation, the regulatory enzymatic machinery controlling these processes and how acetylation works in concert with other regulatory mechanisms (e.g. phosphorylation, other post-translational modifications, lipid composition). That we do not yet address all of these points, we feel does not detract from our manuscript, but rather emphasizes the importance and complexity of the work left to be explored, as our findings will help to direct many future studies in this novel field. From several rounds of reviews we feel that most of the reviewers agree that testing these points is clearly important, but that it is beyond the scope of an already extensive study.

Reviewer #4 (Remarks to the Author):

The authors have satisfactorily addressed all my queries.

REVIEWERS' COMMENTS

Reviewer #2 (Remarks to the Author):

The authors have addressed all my comments. This is an important paper for the field and I hope that it can be published as soon as possible.

Alenka Copic

Reviewer #3 (Remarks to the Author):

The authors have addressed my criticisms in a satisfactory way by quantifying the membrane tubulation and adding discussion that speculates on the differences between lysine acetylation and glutamine mimic. Although dynamic acetylation/deacetylation is a very important question to understand how this modification operates as a regulatory mechanism, I agree that this technically challenging aspect is beyond the scope of the current study. This interesting study is now suitable for publication.